# AVATARSTUDIO: HIGH-FIDELITY AND ANIMATABLE 3D AVATAR CREATION FROM TEXT

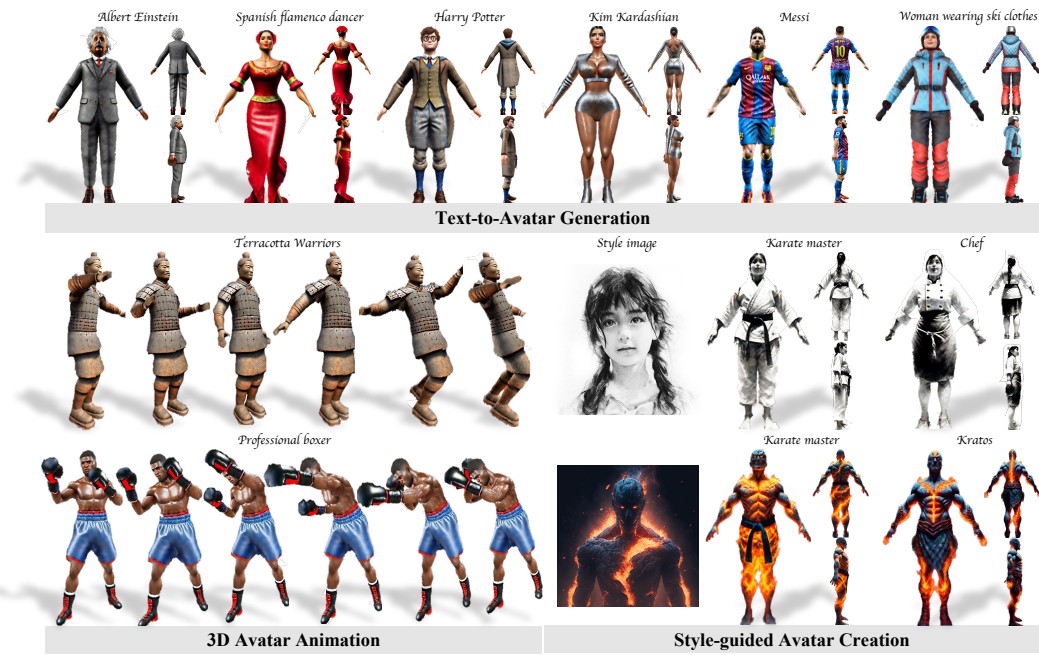

Figure 1: With only text inputs, AvatarStudio generates high-fidelity, animatable 3D avatars featuring realistic textures and detailed geometry, including high-resolution faces and varied clothing styles. A unique feature of AvatarStudio is its easy-to-use animation capability, which allows users to animate the generated avatars via multimodal signals, such as a dancing video or a motion described by text (*e.g.*, "A person is doing boxing"). Furthermore, AvatarStudio supports the creation of avatars with distinct artistic styles (*e.g.*, sketch style) given an additional reference style image.

## ABSTRACT

We study the problem of creating high-fidelity and animatable 3D avatars from only textual descriptions. Existing text-to-avatar methods are either limited to static avatars which cannot be animated or struggle to generate animatable avatars with promising quality and precise pose control. To address these limitations, we propose AvatarStudio, a coarse-to-fine generative model that generates explicit textured 3D meshes for animatable human avatars. Specifically, AvatarStudio begins with a low-resolution NeRF-based representation for coarse generation, followed by incorporating SMPL-guided articulation into the explicit mesh representation to support avatar animation and high-resolution rendering. To ensure view consistency and pose controllability of the resulting avatars, we introduce a 2D diffusion model conditioned on DensePose for Score Distillation Sampling supervision. By effectively leveraging the synergy between the articulated mesh representation and the DensePose-conditional diffusion model, *AvatarStudio* can create high-quality avatars from text that are ready for animation, significantly outperforming previous methods. Moreover, it is competent for many applications, *e.g.*, multimodal avatar animations and style-guided avatar creation. Our project page is https://avatarstudio3d.github.io/.

# 1 INTRODUCTION

The creation of high-fidelity and animatable 3D human avatars is essential in various fields, including media industry, VR/AR, game design, *etc*. However, it is a labor-intensive task that typically requires pre-captured templates and extensive work from experienced artists. Therefore, a user-friendly system that can generate and animate 3D avatars is of great value, which is the primary goal of this work. Existing 3D avatars creation methods can be classified into three categories: (1) template-based generation pipeline (Met, 2023), (2) 3D generative models (Hong et al., 2022a; Zhang et al., 2022) and (3) 2D-lifting methods (Poole et al., 2022; Lin et al., 2023a). Avatars generated using template-based methods typically exhibit relatively simple topology and texture. On the other hand, 3D generative models often struggle to generalize to arbitrary avatars with diverse appearances due to the scarcity and limited diversity of accessible 3D models. Yet, in real-world applications, users often desire high-quality 3D avatars with intricate structures and artistic styles.

Recently, 2D-lifting methods have shown that 2D generation models trained on large-scale image datasets possess strong generalizability, making them suitable for 3D content creation. Representative works such as DreamFusion (Poole et al., 2022) and Magic3D (Lin et al., 2023a), employ 2D diffusion models as supervision to optimize 3D representations using Score Distillation Sampling. More recent studies (Cao et al., 2023; Huang et al., 2023; Kolotouros et al., 2023) incorporate parametric human priors (Loper et al., 2015; Alldieck et al., 2021) into the 2D-lifting optimization process to facilitate 3D human avatar creation. Nevertheless, these methods either focus primarily on creating static avatars, which makes them difficult to animate, or produce low-quality animatable 3D avatars (*e.g.*, blurriness, coarseness, lack of details, and poor pose controllability ) (Cao et al., 2023; Jiang et al., 2023; Hong et al., 2022b; Kolotouros et al., 2023; Huang et al., 2023), which fail to satisfy the requirements for practical applications. Consequently, there is a growing need for more advanced solutions capable of generating high-fidelity, animatable 3D avatars.

In this work, we propose *AvatarStudio*, a novel framework designed for creating high-quality 3D avatars from textual descriptions while offering flexible animation ability. To achieve this, we propose a new 3D human representation that incorporates articulated human modeling into explicit mesh representation. The former enables to animate the generated avatars to desired poses, while the latter allows to fully harness the power of 2D diffusion priors at high-resolution. Specifically, we train a human NeRF from scratch with a pre-defined canonical pose. Taking this learned canonical representation as initialization, we further optimize a SMPL-guided articulated textured avatar mesh represented by DMTet (Shen et al., 2021). The mesh-based representation allows us to render high-resolution images up to $512^2$ through an efficient rasterization-based renderer (Laine et al., 2020), facilitating high-fidelity avatar creation. To improve the animation quality and the pose controllability, we jointly optimize the textured avatar mesh in both deformed and canonical spaces.

To optimize the proposed articulated avatar representation from text, we utilize pre-trained 2D diffusion models as priors. Previous methods typically use StableDiffusion (Rombach et al., 2021) or skeleton-conditional ControlNet (Zhang et al., 2023b) for SDS supervision, which suffer from inaccurate pose control and the Janus problem (Cao et al., 2023; Huang et al., 2023). In contrast, we leverage ControlNet conditioned on DensePose (Güler et al., 2018) as guidance, which offers two benefits: 1) 3D-aware DensePose ensures a more stable and view-consistent avatar creation process; 2) it enables more accurate pose control of the generated avatars.

As shown in Fig. 1, AvatarStudio can create high-fidelity, animatable avatars from text. We evaluate it quantitatively and qualitatively, verifying its superiority over previous state-of-the-arts. Thanks to the **easy-to-use** animation capability, it allows users to animate the generated avatars using multi-modal signals (*e.g.*, video and text). Moreover, by simply plugging an additional adapter (Ye et al., 2023), it is able to create avatars with **unique artistic styles** given a reference style image, further expanding the range of applications and customization options for 3D avatar creation.

# 2 RELATED WORK AND PRELIMINARIES

## 2.1 RELATED WORKS

**Text-guided 3D Content Generation.** The successful advancement in text-guided 2D image generation has paved the way for text-guided 3D content creation. Notable examples include CLIP-

forge (Sanghi et al., 2021), DreamFields (Jain et al., 2021), and CLIP-Mesh (Khalid et al., 2022), which utilize the widely acclaimed CLIP (Radford et al., 2021) to optimize underlying 3D representations, such as NeRF and textured meshes. DreamFusion (Poole et al., 2022) proposes to use the Score Distillation Sampling (SDS) loss, derived from a pre-trained diffusion model (Saharia et al., 2022) as supervision during optimization. Subsequent improvements over DreamFusion include optimizing 3D representations in a latent space (Metzer et al., 2022) and coarse-to-fine manner (Lin et al., 2023a). Furthering this line of research, TEXTure (Richardson et al., 2023) opts to generate texture maps for a given 3D mesh, while ProlificDreamer (Wang et al., 2023) introduces variational score distillation to produce promising results. However, despite these advancements, when it comes to avatar creation, these techniques often exhibit limitations including low quality generation, presence of the Janus problem, and incorrect rendering of body parts. In contrast, AvatarStudio enables high-fidelity and animatable generation of 3D avatars from text prompts.

**Text-guided 3D Avatar Generation.** To enable 3D avatar generation from text, several approaches have been proposed. Avatar-CLIP (Hong et al., 2022b) sets the foundation by initializing human geometry with a shape VAE and utilizing CLIP (Radford et al., 2021) to assist in geometry and texture generation. DreamAvatar (Cao et al., 2023) and AvatarCraft (Jiang et al., 2023) integrate the human parametric model with pre-trained 2D diffusion models for 3D avatar creation. DreamHuman (Kolotouros et al., 2023) further introduces a camera zoom-in technique to refine the local details of resulting avatars; while DreamWaltz (Huang et al., 2023) incorporates a skeleton-conditioned ControlNet and develops an occlusion-aware SDS guidance for pose-aligned supervision. Although these methods achieve animatable results, they suffer from low-quality generation issues, like blurriness, coarseness and insufficient details. Additionally, the weak SDS guidance and the inherent sparsity of skeleton conditioning makes it difficult to generate multi-view consistent avatars with accurate pose controllability. In contrast, we propose an articulated textured mesh representation for 3D human modeling, enabling effective avatar animation and high-resolution rendering. It allows the model to fully utilize 2D diffusion priors at high-resolution, leading to higher-quality generation. Moreover, we use a DensePose-conditioned ControlNet for SDS guidance to ensure more stable, view-consistent avatar creation and improved pose control. A concurrent work, AvatarVerse (Zhang et al., 2023a) also employs DensePose as conditioning for SDS guidance. However, it is limited to static avatar generation, making it hard to animate the resulting avatars in a user-friendly way.

## 2.2 PRELIMINARIES

**Score Distillation Sampling.** The key technique of lifting a pre-trained 2D diffusion model $\epsilon_\phi$ into a 3D representation $\theta$ is the Score Distillation Sampling (SDS), which can be used to guide the generation of 3D content given an input text prompt $y$. Specifically, given a rendered image $\mathcal{I} = g(\theta)$ from a differentiable 3D model $g$, we add random noise $\epsilon$ to obtain a noisy image. The SDS loss then computes the gradient of $\theta$ by minimizing the difference between the predicted noise $\epsilon_\phi(x_t; y, t)$ and the added noise $\epsilon$, which can be formulated as:

$$\nabla_\theta \mathcal{L}_{\text{SDS}}(\phi, x_\theta) = \mathbb{E}_{t,\epsilon}\left[w(t)(\epsilon_\phi(z_t; y, t) - \epsilon)\frac{\partial x}{\partial \theta}\right], \tag{1}$$

where $z_t$ is the noisy image at noise level $t$, $w(t)$ denotes a weighting function depends on $t$ and $y$.

**SMPL.** Skinned Multi-Person Linear model (Loper et al., 2015) is a parametric human model that represents a wide range of human body poses and shapes. It defines a deformable mesh $\mathcal{M}(\xi, \beta) = (\mathcal{V}, \mathcal{S})$, where $\xi$ and $\beta$ denote the pose and shape parameters, $\mathcal{V}$ is the set of $N_v = 6890$ vertices, and $\mathcal{S}$ is the set of linear blend skinning (LBS) weights assigned for each vertex. It provides an articulated geometric proxy to the underlying dynamic human body. In this paper, we develop an articulated 3D human representation for animatable avatar creation by generalizing LBS of SMPL.

## 3 METHODOLOGY

In this work, we aim to generate high-fidelity and animatable 3D human avatar from only text inputs. In Sec. 3.1, we first present how to design an articulated explicit mesh representation for animatable avatar modeling and high-resolution rendering. Then, in Sec. 3.2, we elaborate on how to optimize the proposed representation from text inputs via a DensePose-conditional ControlNet and introduce several simple-yet-effective strategies to facilitate the generation process.

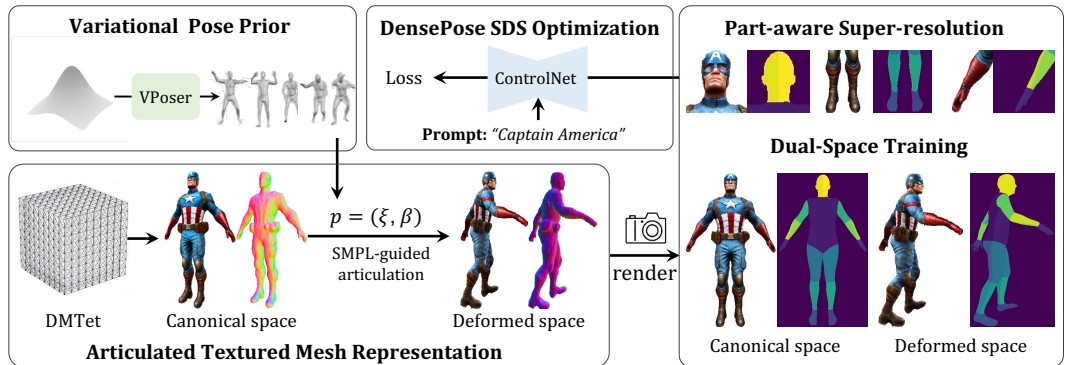

Figure 2: The overview of AvatarStudio. It takes a text prompt as input to optimize an articulated textured mesh representation via a DensePose-conditioned ControlNet for high-quality and animatable 3D avatar creation. To facilitate the optimization process, it leverages several simple-yet-effective strategies, like part-aware super-resolution and dual-space training. See main text for more details.

## 3.1 ARTICULATED 3D HUMAN MODELING

**SMPL-guided Avatar Articulation.** To create animatable human avatars, we incorporate SMPL-guided articulation into the 3D human modeling process to drive the generated avatar to the desired poses. Specifically, given SMPL parameter $p = (\xi, \beta)$, our model first generates a template avatar with a pre-defined pose in the *canonical space*, followed by deforming it to the target pose defined by the corresponding parameter $p$ in the *deformed space*. We leverage the inverse transformation of SMPL LBS to guide the deformation of our human representation. Specifically, given a point $\mathbf{x_d}$ in the *deformed space*, we first find its nearest vertex $v^*$ in the corresponding SMPL mesh, and then use the skinning weights of $v^*$ to deform $\mathbf{x_d}$ to the corresponding point $\mathbf{x_c}$ in the *canonical space*:

$$\mathbf{x_c} = \mathcal{G}^{-1} \cdot \mathbf{x_d}, \quad \mathcal{G} = \sum_{i=1}^{N_j} s_i^* \cdot B_i(\xi, \beta) \tag{2}$$

where $s_i^*$ is the skinning weight of vertex $v^*$ *w.r.t* the $i$-th joint, $B_i(\xi, \beta)$ is the bone transformation matrix of joint $i$, $N_j = 24$ is the number of joints.

**Articulated Textured Mesh Representation.** Most existing text-to-avatar methods (Jiang et al., 2023; Cao et al., 2023) represent human avatars as NeRF and use volumetric rendering for avatar image rendering. However, these methods typically require a considerable computation budget, limiting the resolution of generated images. As a result, they often produce low-quality avatars with coarseness and lack of detail. In contrast, we opt for an explicit mesh representation for animatable human modeling, which enables high-resolution rendering via a highly efficient rasterizer (Laine et al., 2020). Nevertheless, we empirically observe that directly optimizing the explicit mesh representation for avatar generation produces degenerated results due to the high dimensionality of the mesh space and the complexity of human bodies.

To address this, we propose a coarse-to-fine pipeline to optimize the 3D human representation in a progressive manner. In the coarse stage, we adopt NeRF to learn a static human in the *canonical space* by leveraging the low-resolution diffusion prior as guidance. We use hash grid decoding from InstantNGP (Müller et al., 2022) with a two-layer MLP to predict the density and color. To ease the learning difficulties, we adopt a residual prediction scheme on top of the SMPL-derived density field, which serves as a strong geometric prior. For more details, please refer to the Appendix.

In the fine stage, we use a differentiable surface representation, *i.e.*, Deep Marching Tetrahedra (DMTet) (Shen et al., 2021), to model avatars as textured meshes. The explicit mesh representation allows us to improve the generation quality by optimizing with high-resolution diffusion prior (*e.g.*, $512 \times 512$). Particularly, DMTet represents the surface of humans with a discrete signed distance field defined on a deformable tetrahedral grid, where a mesh face will be extracted if two vertices of an edge in a tetrahedron have different signs of SDF values. To inherit the learned geometry prior from the previous stage, we initialize DMTet with the mesh extracted from the coarse NeRF using the marching cube algorithm (Lorensen & Cline, 1998).

For articulating avatar modeling, we establish the correspondence between the *canonical* and *deformed spaces* via the SMPL-guided deformation. In specific, for a point $x_d$ in the *deformed space*, we first find the corresponding point $x_c$ in the *canonical space* (see Eq. 2). We then predict a signed distance offset from the surface of the mesh extracted from the coarse model for geometry refinement. The final signed distance of the fine stage $d_{fine}(\mathbf{x_c})$ at point $x_d$ can be computed as:

$$d_{fine}(\mathbf{x_c}) = d_{coarse}(\mathbf{x_c}) + \Delta d(x_c), \tag{3}$$

where $d_{coarse}(\mathbf{x_c})$ is the signed distance value from the coarse stage, $\Delta d(x_c)$ is the residual SDF value predicted by a two-layer MLP. This allows us to animate the generated avatars to arbitrary poses by simply deforming the canonical one. We employ the neural color field initialized from the coarse stage for mesh textures modeling under higher-resolution space.

## 3.2 TEXT-TO-AVATAR GENERATION

**DensePose SDS Optimization.** We employ 2D diffusion models as priors for guiding the 3D human generation process. The core idea is to optimize the 3D model by distilling prior knowledge from a pretrained diffusion model using Score Distillation Sampling (SDS) loss. Although the image diffusion model can guide content generation, it struggles to synthesize a human avatar with the correct pose due to the absence of conditioning signals. To address this, some previous methods (Huang et al., 2023) utilize skeleton-conditioned ControlNet for SDS supervision. However, they still suffer from inaccurate pose control and the Janus problem due to the sparsity of keypoints signal. In this work, we adopt a DensePose-conditioned ControlNet that leverages more expressive DensePose signal as condition for avatar generation. Given the SMPL parameter $p$, we render the human image $\mathcal{I} = g(\theta, p)$ from the 3D human model $g$ parametrized by $\theta$. We also render the SMPL mesh defined by $p$ as DensePose conditions $\mathcal{I}_{cond}$ from the same camera viewpoint as $\mathcal{I}$. The DensePose-conditioned SDS loss can be defined as follows:

$$\nabla_\theta \mathcal{L}_{SDS}(\phi, \mathcal{I} = g(\theta, p)) = \mathbb{E}_{t,\epsilon}\left[\omega(t)(\hat{\epsilon}_\phi(\mathcal{I}_t; y, \mathcal{I}_{cond}, t) - \epsilon)\frac{\partial \mathcal{I}}{\partial \eta}\right], \tag{4}$$

where $\mathcal{I}_t$ denotes the noisy image at noise level $t$, $\omega(t)$ is a weighting function that depends on the noise level $t$, $\epsilon$ is the added noise, and $y$ is the input text prompt. Compared to skeleton-conditioned ControlNet, DensePose-conditioned ControlNet offers two benefits: 1) 3D-aware DensePose ensures a more stable and view-consistent avatar creation process; 2) it enables more accurate pose control of the generated avatars.

**Part-aware Super-resolution.** Directly generating the full-body avatars often produce results that are blurry and lack of fine details. To improve the fidelity of the generated avatars, we develop a part-level super-resolution strategy for both the coarse and fine stages. By leveraging the body prior from SMPL, we can easily determine the positions of different body parts (*i.e.*, head, hand, upper body, lower body, and arm). We zoom in on each part and apply SDS as before to refine their texture and geometric details. To guide this fine-grained optimization, we use the corresponding text prompts for each body part (*e.g.*, "The headshot of <name>", "The right hand of <name>", etc), where <name> is the textual description of an avatar.

**Dual Space Training.** To improve the quality of animation while maintaining high-quality textures and geometries, we adopt a dual-space training strategy that jointly optimizes the human avatar in both the *canonical space* and *deformed space*. We utilize "A-pose" in the *canonical space* as it is a common pose for natural humans. Within the *deformed space*, we sample different poses for training to enhance pose control generalization and accuracy. In particular, we randomly sample human poses from VPoser (Pavlakos et al., 2019), a variational autoencoder that learns a latent representation of the human pose prior, during the training process.

**CFG Rescale.** To ensure better alignment with input text, existing works often use a large classifier-free guidance (CFG) scale when optimizing avatar representation with SDS. However, it comes at a cost of causing severe color saturation, making the generated avatars look unreal. To alleviate the color saturation issue, we apply the CFG rescale trick from Lin et al. (2023b) for adjusting the denoised $\hat{x}_0$. We encourage the readers to refer to Lin et al. (2023b) for more details.

## 4 EXPERIMENTS

In this section, we first verify AvatarStudio's ability for 3D avatar creation from text inputs. Then, we conduct ablation studies to analyze the effectiveness of each component. Finally, we showcase the applications of AvatarStudio, including multimodal avatar animation and style-guided creation.

**Implementation Details.** To train DensePose-conditioned ControlNet, we sample human images from the LAION dataset (Schuhmann et al., 2022) and annotate them using a pre-trained DensePose model (Güler et al., 2018), resulting in around 1.2M image pairs. The ControlNet training is based on the Stable Diffusion v2.1 base model ($512^2$) and takes about 2 days using 16 NVIDIA V100 GPUs. Our AvatarStudio is implemented in the threestudio (Guo et al., 2023) codebase. For each text prompt, AvatarStudio trains the 3D model with a batch size of 1 for 10k and 3k iterations in the coarse and fine stages, respectively, using the AdamW optimizer (Kingma & Ba, 2015) at a learning rate of 0.01. The entire training process takes around 2.5 hours on a single NVIDIA V100 GPU. For the SDS guidance, the maximum and minimum timestep decrease from 0.98 to 0.5 and 0.02, over the first 8,000 steps in the coarse stage. In the fine stage, these are fixed to 0.5 and 0.02, respectively. We set rescale factor to 0.5 for the CFG rescale trick. The rendering resolution begins at $64^2$ and increases to $256^2$ after the first 5,000 steps in the coarse stage and is set to $512^2$ in the fine stage. For more implementation details, please refer to Appendix.

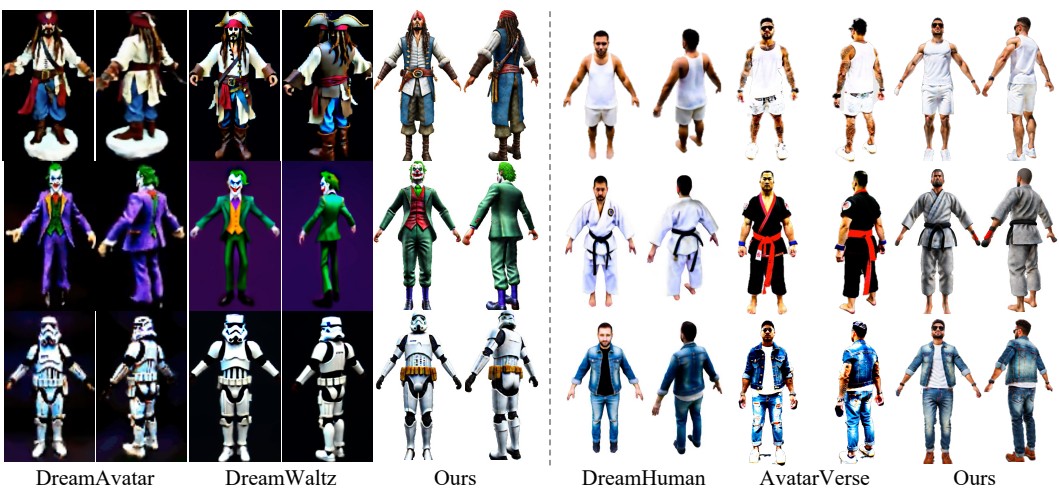

| DreamAvatar | DreamWaltz | Ours | DreamHuman | AvatarVerse | Ours |

Figure 3: Qualitative comparisons with four SOTA methods. AvatarStudio generates more realistic and higher-resolution avatars with fine-grained geometries, like cloth wrinkles, compared with other methods. The prompts we used for comparisons: $1^{st}$ row: "A standing Captain Jack Sparrow from Pirates of the Caribbean"; "A man wearing a white tanktop and shorts", $2^{nd}$ row: "Joker"; "A karate master wearing a Black belt", $3^{rd}$ row: "Stormtrooper"; "A man wearing a jean jacket and jean trousers". Best viewed in $2\times$ zoom. For more results, please refer to our appendix and project page.

### 4.1 QUALITATIVE COMPARISON

We present a qualitative comparison against DreamAvatar (Cao et al., 2023), DreamWaltz (Huang et al., 2023), DreamHuman (Kolotouros et al., 2023) and AvatarVerse (Zhang et al., 2023a) in Fig. 3. These methods, like ours, also employ human priors and 2D diffusion models for the creation of 3D avatars. Benefiting from the explicit mesh representation, AvatarStudio outperforms the Dream* methods significantly in terms of both geometry and texture, resulting in richer details across all cases. In comparison with AvatarVerse, our AvatarStudio generates avatars with clearer appearances ($1^{st}$ and $3^{rd}$ rows) and align more closely with the input texts ($2^{nd}$ row). Moreover, thanks to its articulation modeling, a standout feature of AvatarStudio is its ability to support avatar animation (see later in Fig. 7), which is not available in AvatarVerse. These clearly demonstrate the superiority of AvatarStudio for text-guided 3D avatar creation. We also visualize the normal maps of the generated avatars in Fig. 4 and show that our method produces high-quality and detailed geometry.

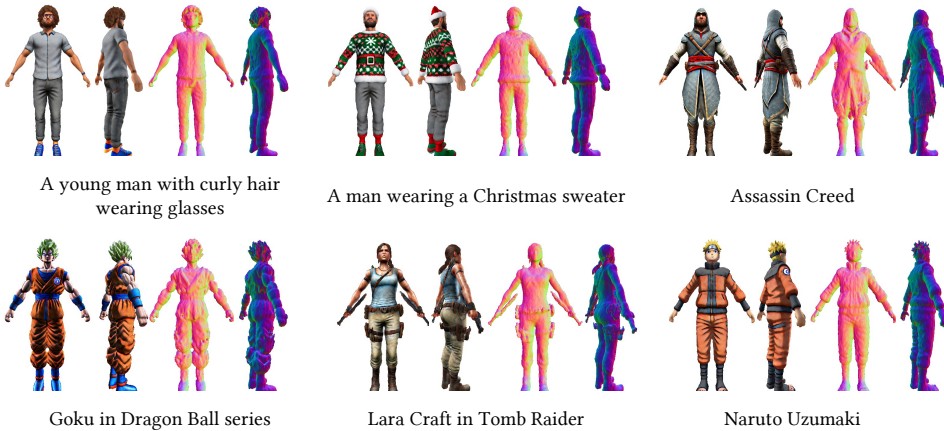

A young man with curly hair wearing glasses

A man wearing a Christmas sweater

Assassin Creed

Goku in Dragon Ball series

Lara Craft in Tomb Raider

Naruto Uzumaki

Figure 4: Our AvatarStudio produces high-quality and detailed geometry.

## 4.2 QUANTITATIVE EVALUATION

**User Study.** To quantitatively evaluate AvatarStudio, we conduct user studies comparing the performance of our results with four SOTA methods under the same text prompts. We randomly pick 30 samples to conduct user studies. Each text prompt is evaluated by 20 volunteers. In Fig. 5, we first compare AvatarStudio with DreamAvatar (Cao et al., 2023) and DreamWaltz (Huang et al., 2023) for specific characters generation, and then compare with AvatarVerse and DreamHuman (Kolotouros et al., 2023) in terms of realistic humans generation. The results demonstrate that our method achieves significantly superior preference over all other methods.

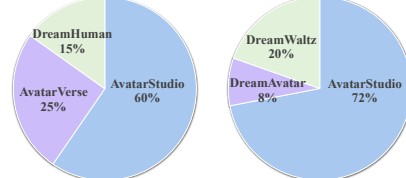

Figure 5: User preference.

**CLIP Score.** We use CLIP score, implemented by `TorchMetrics` (tor, 2023), as an evaluation metric to measure the consistency between the generated avatars and input texts for the above methods. For each method, we render its generated avatars from four evenly distributed horizontal views, and calculate the averaged CLIP score for these rendered images and the input text. Similar to the user study, we compare with DreamAvatar and DreamWaltz in term of specific characters generation, and compare with AvatarVerse and DreamHuman for realistic human generation. The CLIP scores for DreamAvatar, DreamWaltz and ours are 30.45, 31.52 and 32.80, respectively, while the CLIP score for DreamHuman, AvatarVerse and ours are 29.54, 28.88 and 32.17, respectively. Our AvatarStudio consistently outperforms all these methods, verifying its effectiveness in creating more accurate avatars in alignment with the input texts.

## 4.3 ABLATION STUDIES

**Avatar Representation.** Our approach, AvatarStudio, utilizes an articulated mesh representation in a coarse-to-fine manner, with the coarse stage being represented by NeRF. To explore the impact of different 3D representations, we optimize 3D avatars from text using either mesh-only (DMTet) or NeRF-only representations. As shown in Fig. 6 (a), directly optimizing meshes for avatar creation results in collapsed results, while using NeRF-only representation often yields avatars of lower quality. In contrast, our proposed articulated representation, which combines NeRF and mesh, successfully generates high-resolution images with fine details, demonstrating its effectiveness.

**DensePose-conditioned ControlNet.** AvatarStudio uses ControlNet conditioned on DensePose for SDS guidance. To assess its efficacy, we compare the performance of our method when trained with StableDiffusion (SD) or Skeleton-conditioned ControlNet (see Fig. 6 (b)). We observe the model guided by StableDiffusion generates avatars that exhibit incorrect poses and lower quality due to the lack of pose-aware guidance, which results in inaccurate animations. While the Skeleton-conditioned ControlNet model improves pose control, it still suffers from inaccuracies in foot positioning and head orientation. In contrast, our proposed DensePose-conditioned diffusion guidance

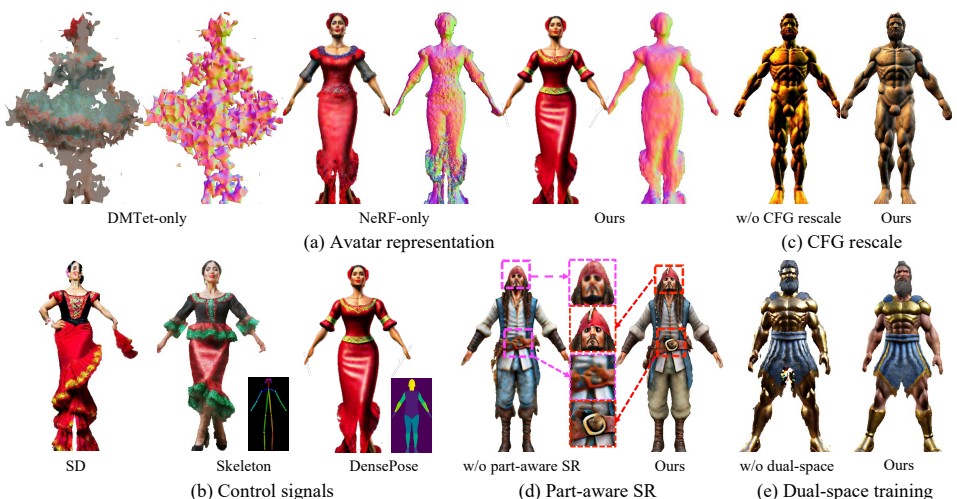

DMTet-only · NeRF-only · Ours · w/o CFG rescale · Ours
(a) Avatar representation · (c) CFG rescale

SD · Skeleton · DensePose · w/o part-aware SR · Ours · w/o dual-space · Ours
(b) Control signals · (d) Part-aware SR · (e) Dual-space training

Figure 6: Ablation studies on different components.

achieves precise and stable pose control, accompanied by high-quality textures, which validates the importance of leveraging DensePose-conditioned guidance in the avatar creation process.

To quantitatively assess the pose controllability of avatars generated with different diffusion guidances, we predict the SMPL parameters for posed avatar images using a pre-trained 3D human reconstruction model HybrIK (Li et al., 2021). There images are generated via given SMPL parameters. We calculate the Mean Squared Error (MSE) in $10^{-2}$ between the input and the estimated SMPL parameters. Specifically, for each avatar, we generate 120 posed images using 120 fixed SMPL parameters in a frontal view. We compute the average MSE across those images as the final MSE score. The MSE for StableDiffusion, Skeleton-conditioned ControlNet and our method are 9.0, 7.7, 5.9, respectively. AvatarStudio achieves the best pose control with the lowest MSE, further verifying the effectiveness of DensePose guidance.

**Part-aware Super-resolution and CFG Rescale Strategy.** Furthermore, we explore the individual impacts of part-aware super-resolution (SR) and CFG rescale strategy. As shown in Fig. 6 (c), we observe CFG rescale method can mitigate the color saturation issue, generating more natural appearance for the generated avatar. Upon the addition of part-aware super-resolution, the model can produce sharper appearances and more local fine details, such as on faces and belts (see Fig. 6 (d)). These studies validate the effectiveness of each proposed component in our approach, demonstrating their substantial contribution to the final result.

**Dual-space Training.** To validate the effectiveness of the dual-space training, we compare with a baseline that trains on canonical space only. For quantitative evaluations, we report the pose controllability score by computing the MSE in $10^{-2}$ between the input and the estimated parameters: 6.5 and 5.9 for canonical space only and dual-space training, respectively. We further visualize the generated RGB images for qualitative comparison (see Fig. 6 (e)). We can see that, without dual-space training, the generated avatar exhibits poor details when deformed to a different pose, suggesting that dual-space training is essential to improve the robustness against different poses.

## 4.4 APPLICATIONS

**Multimodal Animation.** A crucial feature of our method lies in its capability to provide high-quality, natural and easy-to-use animation, which allows users to drive avatars using multimodal signals (*e.g.*, video, text, audio, *etc*). Fig. 7 shows the animation of avatars created by AvatarStudio using either video (a) or text (b). For video-driven animation, we first use VIBE (Kocabas et al., 2021) to estimate SMPL sequences from the driving video, which are then leveraged to animate the generated avatar. For text-driven animation, we adopt MDM (Tevet et al., 2023) to convert text into SMPL sequences. Benefiting from the articulation modeling integrated into our explicit mesh representation, the generated avatars can be easily animated, exhibiting natural movements. The consistency of these results *w.r.t.* SMPL motions ensures that avatars generated by AvatarStudio can leverage any multimodal-to-motion methods that output SMPL sequences for animation. These

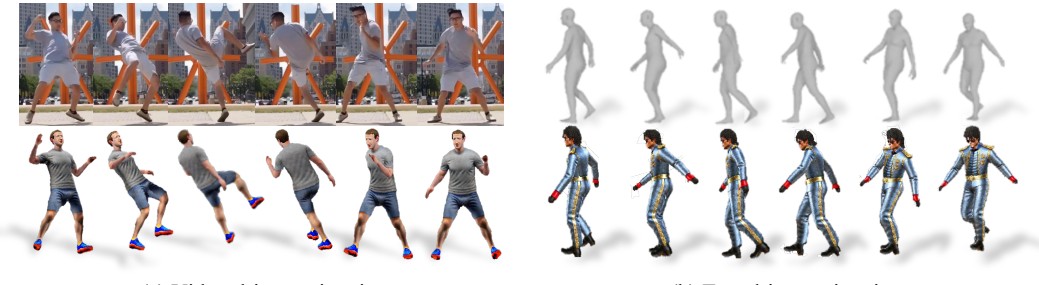

| (a) Video-driven animation | (b) Test-driven animation |

Figure 7: AvatarStudio facilitates the animation of avatars using multimodal signals. We demonstrate examples of animated avatars, *i.e.*, "Mark Zuckerberg" and "Michael Jackson", using video-driven motion and text-driven motion ("A person is doing Moonwalk"), respectively.

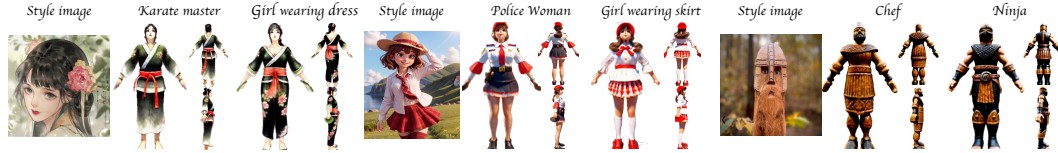

Figure 8: Our AvatarStudio also supports style-guided avatar creation by simply providing an additional style image. Note that the provided style image can be combined with text prompt to enable flexible avatar creation (*e.g.*, a police woman in Pixar Disney style in the middle).

examples showcase the versatility and potential of our method in creating realistically animated avatars from diverse text prompts. For more results, please refer to our project page.

**Style-guided Avatar Creation.** Moreover, we show that AvatarStudio supports stylized avatar creation by simply providing an additional style image. To achieve this, we employ IP-Adapter (Ye et al., 2023), an adapter that enables image prompt capability for pre-trained text-to-image diffusion model via a decoupled cross-attention design. We plug IP-Adapter into our DensePose-conditioned ControlNet and optimize with SDS as before. Without bells and whistles, AvatarStudio can generate high-quality avatars of various styles of interests as shown in Fig. 8. Note that the provided style image can be combined with text prompt to enable flexible avatar creation (*e.g.*, a police woman in Pixar Disney style in the middle of Fig. 8). This capability expands its application, allowing users to create stylized avatars catering to specific aesthetic desires.

## 5 DISCUSSION AND CONCLUSION

In this paper, we introduce AvatarStudio for creating high-fidelity and animatable 3D avatars from only textual inputs. In short, AvatarStudio introduces articulated modeling into explicit 3D mesh representation to support avatars animation while offering high rendering quality. To further improve pose contrallability and view consistency, we leverage DensePose-conditioned ControlNet for Score Distillation Sampling supervision. We also discover several simple yet effective strategies, such as part-aware super-resolution for improving the fidelity of each body part, dual-space training for improving the robustness against different poses and CFG rescale to alleviate the color saturation issue. As a result, AvatarStudio supports various downstream applications, including multimodal avatar animations (*e.g.*, video or text driven) and style-guided avatar creation.

**Limitations.** While AvatarStudio can generate high quality, animatable 3D avatars, there are still rooms for improvement: 1) Although AvatarStudio supports animation of the generated avatars, it currently does not support fine-grained motions, such as changes in facial expressions. 2) The efficiency of AvatarStudio could be improved as it currently takes about 2.5 hours to generate a 3D avatar from a text input. 3) Artifacts, particularly in the avatars' hands, can occasionally be observed.

**Social Impact.** The capabilities of 3D avatar creation, animation, and image-guided avatar stylization could be misused for generating fake videos or manipulating authentic videos for illicit purposes. Such potentially harmful applications could pose sizable societal threats. We strictly forbid these kinds of abuses of our technology. Moreover, AvatarStudio might exhibit bias in avatar generation due to an unbalanced distribution within the training data used for the 2D diffusion model.

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

# A    APPENDIX

In this Appendix, we provide additional implementation details in Section A.1, additional baseline comparisons and more visual results in Section A.2. We discuss limitations and future work in Section A.3. Please refer to our project page: https://avatarstudio3d.github.io/ for results in video format.

## A.1    IMPLEMENTATION DETAILS

**Training Details.** We represent the camera position in the world coordinate system by radius, elevation angle, azimuth angle, and field of view (FOV). Here we set the camera distance within the range of [1.0, 2.0], the elevation angle within the range of $[-10°, 60°]$, the azimuth angle within the range of $[0°, 360°]$, and the FOV within the range of [55, 65], respectively. During training, we perform part-aware super-resolution every 3 step. In practice, we empirically find the following prompt postfixes can improve the quality and realism of generated images: "`high quality, 8k uhd, realistic`". We also use "`lowres, bad anatomy, bad hands, missing fingers, worst quality, nsfw`" as negative prompts to improve the generation. Another challenge is the infamous multi-face Janus issue that is due to the lack of 3D awareness of 2D diffusion models. The text prompts often describe the identity and the appearance of the desired human avatar while ignoring the view information. Consequently, the model may generate redundant contents of the character, *i.e.*, face, in other views. To address this, we append view-dependent text to the provided input text prompt according to the randomly sampled camera poses. Specifically, we divide the horizontal angles to "`front view`", "`back view`", and "`side view`", and append "`overhead view`" for camera poses with high elevation angles, following DreamFusion (Poole et al., 2022).

**Coarse Stage NeRF.** In the coarse stage, we adopt NeRF to learn a static human in the *canonical space* by leveraging the low-resolution diffusion prior as guidance. We use hash grid decoding from InstantNGP (Müller et al., 2022) with a two-layer MLP to predict the density and color. The hash grid encoding from Instant NGP allows us to encode high-frequency details at a relative low computational cost. The normal is calculated as the spatial gradient of the density fields. We find that directly learning the geometry of the human from scratch is non-trivial because the human body has complex pose and shape variations. Therefore, we adopt a residual prediction scheme by leveraging the SMPL model as a strong geometry prior. In other words, instead of directly predicting the density value from scratch, we predict a residual density based on the signed distance value from the surface of the SMPL template (Yifan et al., 2021). Specifically, for a point $x_c$ in the *canonical space*, we query the body SDF value $d(x_c)$ to the surface of SMPL, and then convert SDF to the density by:

$$\sigma(\mathbf{x_c}) = \text{softmax}^{-1}(\tau) + \Delta\sigma(x_c), \qquad \tau = \frac{1}{\alpha}\text{sigmoid}(-\frac{d(x_c)}{\alpha}) \qquad (5)$$

where $\Delta\sigma(x_c)$ is the residual density term predicted by the MLP network, $\alpha$ is set to 0.001 in experiments. The residual densities prediction scheme based on the coarse SMPL body mesh can largely alleviate the geometry learning difficulties, yielding better generation results. In practice, we also use grid pruning to reduce the memory cost and speed up the training process. During training, we maintain an occupancy grid and gradually prune sample points in the empty space.

**Background Modeling.** We model the foreground avatars and backgrounds separately. We adopt a neural environment map network similar to DreamFusion (Poole et al., 2022) to model the learnable background. Specifically, we use a two-layer MLP which takes the ray direction position encoding as input and predicts the color. We generate the final image by performing alpha composition (Schwarz et al., 2022):

$$\mathcal{I}_{final} = \mathcal{I}_{fg} + (1 - \mathcal{M}) \cdot \mathcal{I}_{bg}, \qquad (6)$$

where $\mathcal{I}_{fg}$ is the rendered foreground human image, and $\mathcal{I}_{bg}$ is the rendered background image, and $\mathcal{M}$ is the foreground object mask. For the coarse stage, the foreground mask of the NeRF model can be generated by accumulating to density along the ray:

$$\mathcal{M} = \sum_{i=1}^{N} T_i(1 - \exp(-\sigma_i\delta_i)), T_i = \exp(-\sum_{j=1}^{i-1}\sigma_j\delta_j), \qquad (7)$$

where $N$ is the number of samples in each camera ray, $\delta_i$ is the distance between adjacent sample points $i$-th and $(i + 1)$-th, $\sigma_i$ is the volume density of sample $i$. For more details, please refer to NeRF (Mildenhall et al., 2020). In the fine stage, we can directly generate the foreground mask of the textured mesh with the differentiable rasterizer Nvdiffrast (Laine et al., 2020).

**Evaluation Metrics.** We use CLIP score, implemented by `TorchMetrics` (Detlefsen et al., 2022), as an evaluation metric to measure the consistency between the generated avatars and input texts. For each method, we render its generated avatars from four evenly distributed horizontal views, *i.e.*, $[0°, 90°, 180°, 270°]$ and calculate the averaged CLIP score for these rendered images and the input text. Specifically, we use CLIP model "clip-vit-base-patch16" for clip score evaluation.

In order to evaluate the pose controllability of the generated avatars, we predict the SMPL parameters for posed avatar images using a pre-trained 3D human reconstruction model HybrIK (Li et al., 2021). There images are generated via given SMPL parameters. We then calculate the Mean Squared Error (MSE) in $10^{-2}$ units between the input and the estimated SMPL parameters. This MSE provides a quantitative measure of how closely the pose of the generated avatars align with the expected poses, as determined by the input SMPL parameters. Specifically, for each avatar, we generate 120 posed images using 120 fixed SMPL parameters in a frontal view. We compute the average MSE across those images as the final MSE score.

**Loss Functions.** We use the Score Distillation Sampling (SDS) loss (Poole et al., 2022) $\mathcal{L}_{SDS}$ to guide the optimization of 3D human generation. To encourage smooth geometry, we also introduce several regularization terms, including normal smooth loss (Nealen et al., 2006) $\mathcal{L}_c$ and mesh Laplacian smoothing loss (Desbrun et al., 2023) $\mathcal{L}_s$. The overall loss function is formulated as:

$$\mathcal{L}_{total} = \lambda_{SDS}\mathcal{L}_{SDS} + \lambda_c\mathcal{L}_c + \lambda_s\mathcal{L}_s \tag{8}$$

where $\lambda_{SDS} = 1$, $\lambda_c = 10,000$ and $\lambda_s = 10,000$ are weights for the SDS loss $\mathcal{L}_{SDS}$, normal smooth loss $\mathcal{L}_c$ and mesh Laplacian smoothing loss $\mathcal{L}_s$, respectively.

## A.2 ADDITIONAL RESULTS

**More Comparisons.** In fig. S9, fig. S10 and fig. S11, we provide more comparison results with SOTA methods DreamHuman (Kolotouros et al., 2023), AvatarVerse Zhang et al. (2023a) and TADA (Liao et al., 2023), respectively.

**Additional Results of AvatarStudio.** In fig. S12 and fig. S13, we visualize more results from AvatarStudio.

## A.3 LIMITATIONS AND FUTURE WORK

While AvatarStudio displays potential in generating high-quality, there are still rooms for improvement: Firstly, although AvatarStudio supports animation of the generated avatars, it currently does not support fine-grained motions, such as changes in facial expressions. Future improvements can delve into utilizing more expressive parametric human models such as the SMPL-X (Pavlakos et al., 2019) model for avatar creation guidance, which can yield greater expressiveness in avatar animation. Second, the efficiency of AvatarStudio can be further improved. It currently requires approximately 2.5 hours to generate a single 3D avatar from a text input. Toward increasing optimization efficiency, we can consider two promising solutions: 1) adopt-

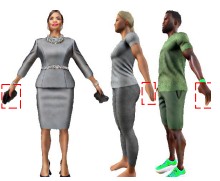

Figure S14: Failure cases.

ing more efficient 3D representations like Gaussian Splatting (Kerbl et al., 2023) for human avatar representation. 2) incorporating more powerful guidance such as Multiview Diffusion (Shi et al., 2023) to expedite the optimization process. Finally, artifacts, particularly in the avatars' hands, can occasionally be observed (see Fig. S14). A potential improvement could lie in the use of a hybrid guidance solution. For example, using DensePose-conditioned ControlNet for body guidance and Skeleton-conditioned ControlNet for hand guidance can mitigate these artifacts.

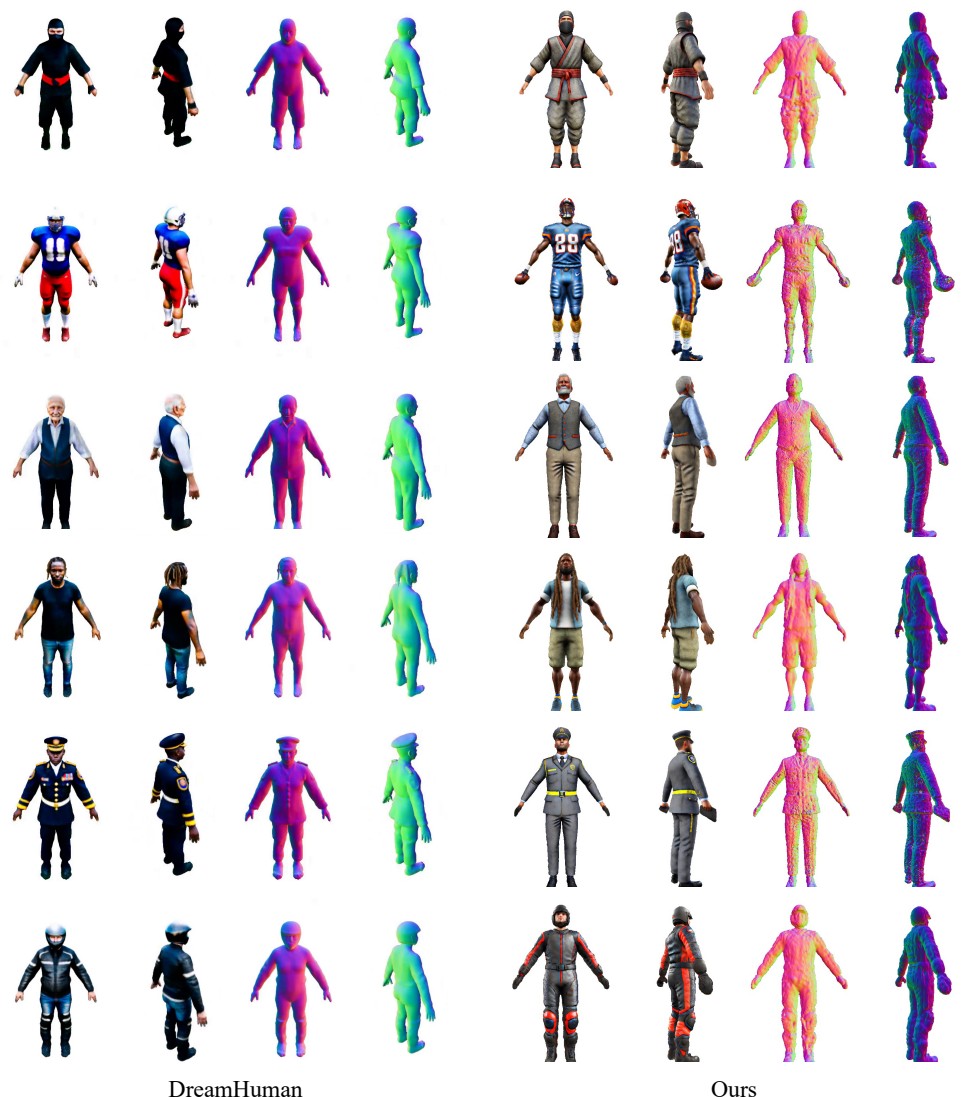

DreamHuman                                              Ours

Figure S9: Visual comparison between AvatarStudio and DreamHuman (Kolotouros et al., 2023).

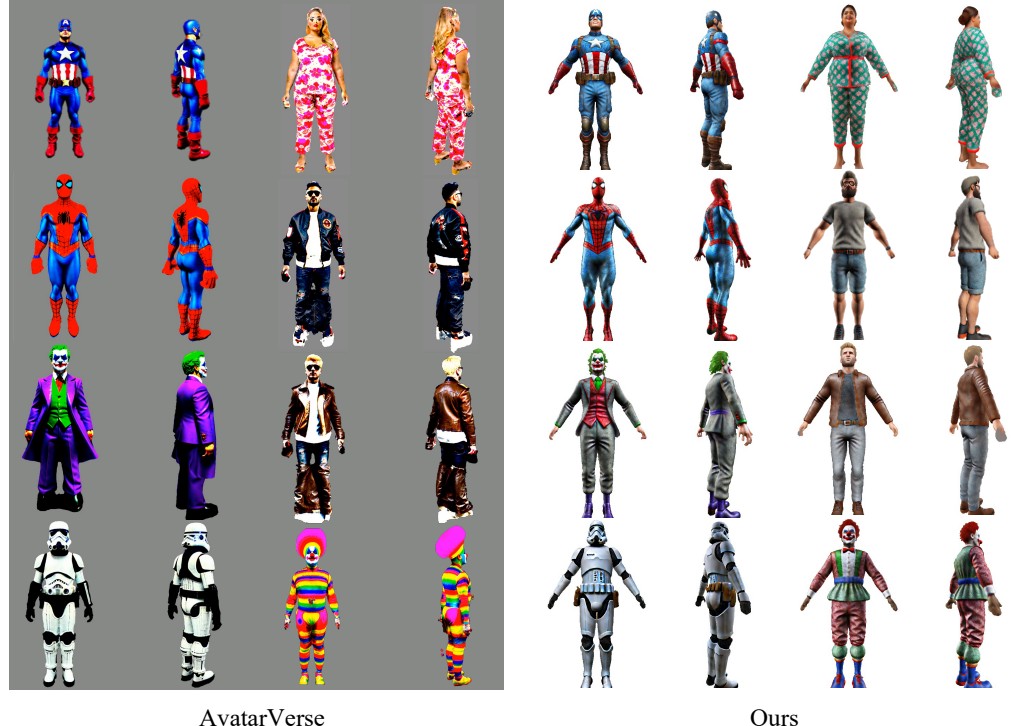

AvatarVerse                    Ours

Figure S10: Visual comparison between AvatarStudio and AvatarVerse (Zhang et al., 2023a).

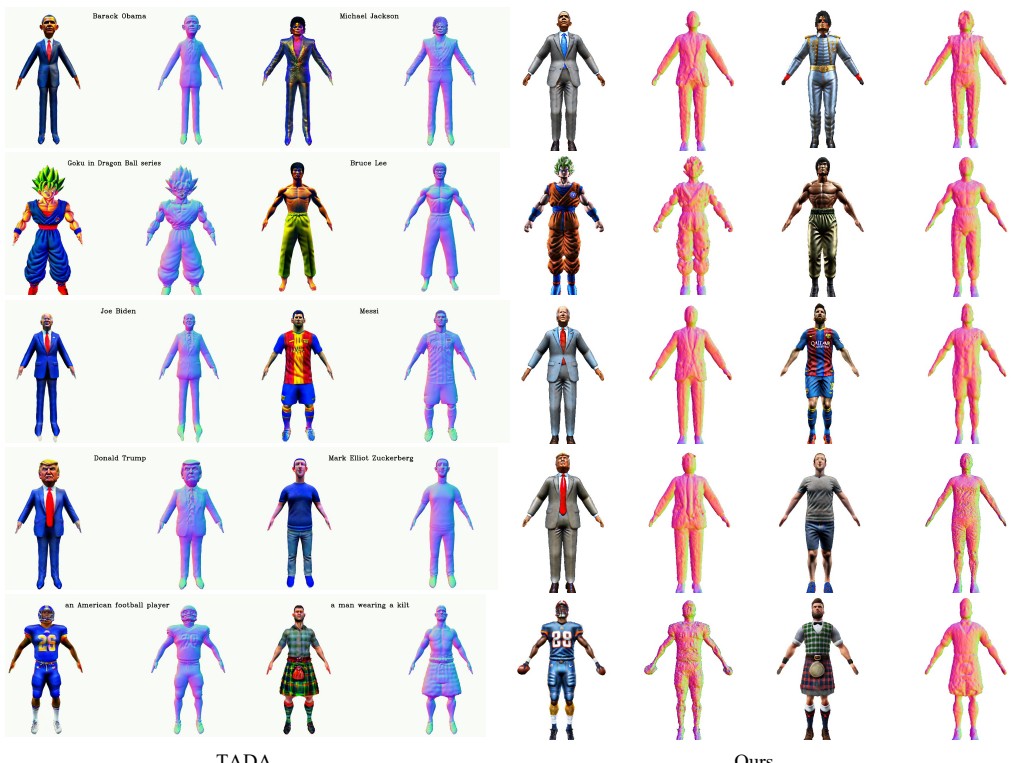

TADA                          Ours

Figure S11: Visual comparison between AvatarStudio and the latest work TADA (Liao et al., 2023).

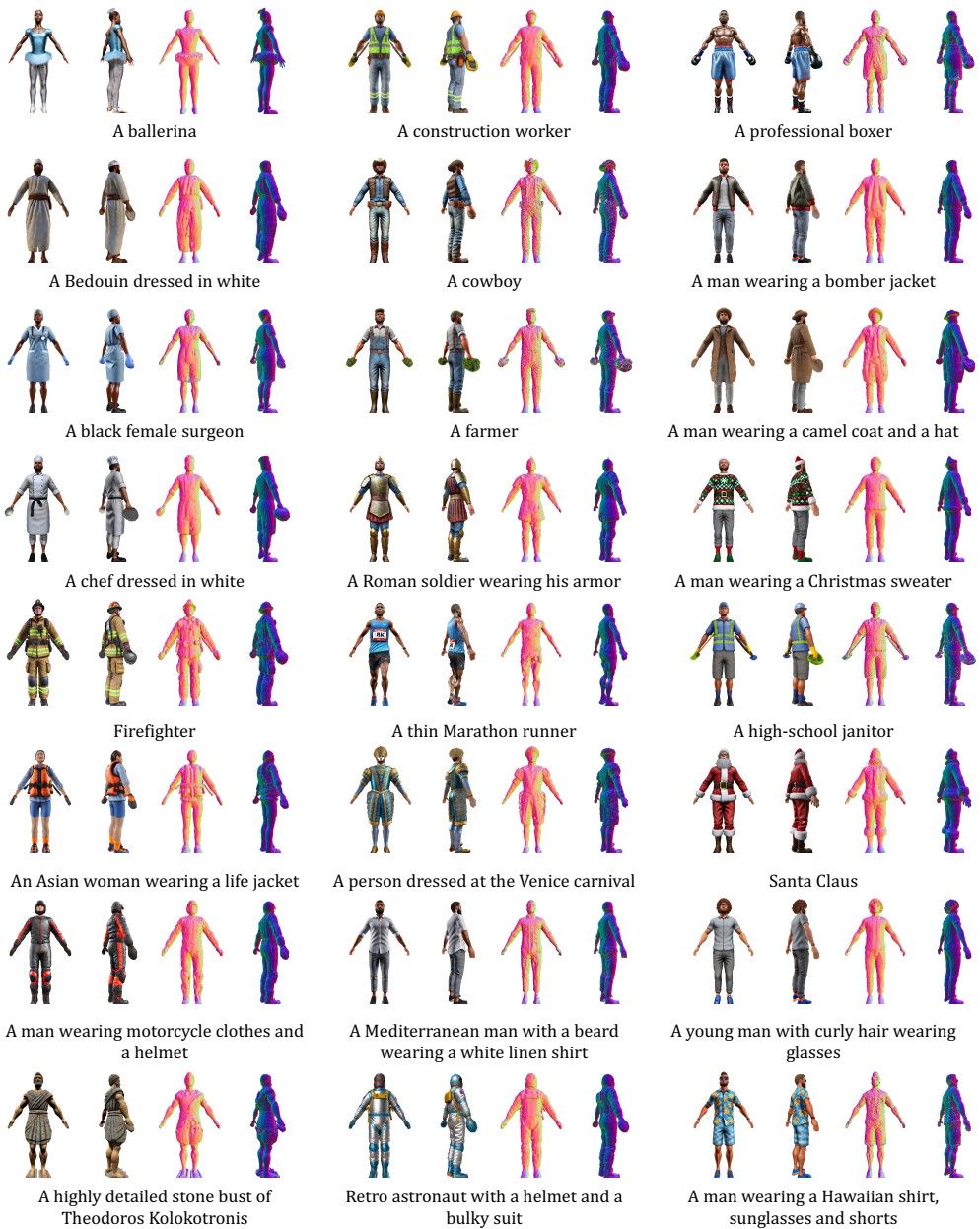

Figure S12: More qualitative results of AvatarStudio.

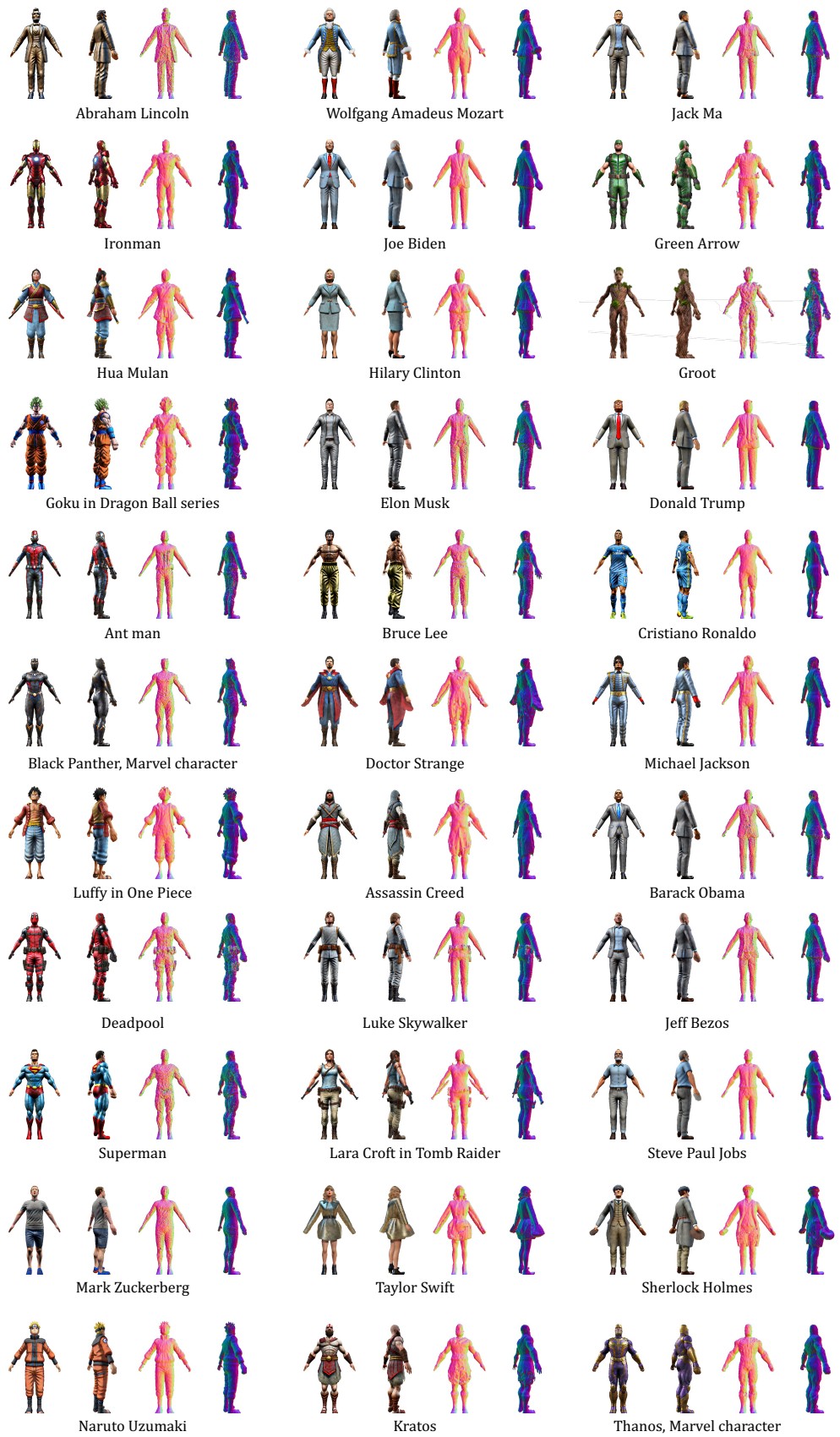

Figure S13: More qualitative results of AvatarStudio.

