# OpenReview forum: "AvatarStudio: High-fidelity and Animatable 3D Avatar Creation from Text"
_ICLR.cc/2024/Conference — Submitted to ICLR 2024_

### Official Review · Reviewer_q3fv · 2023-10-28

**Soundness:** 2 fair
**Presentation:** 1 poor
**Contribution:** 2 fair
**Rating:** 5
**Confidence:** 4

**Summary:**

This paper proposes a pipeline for creating 3D avatar from text prompts. In contrast to the existing work, this method can animate the avatar using the underlying SMPL structure. This is a coarse to fine pipeline where in the coarse stage a residual prediction scheme over SMPL using NERF is proposed which is used to initiaise in the fine stage. In the fine stage DMTet (deep Marching tetrahedra) is used create a high resolution avatar. Further there is a part aware superresolution. For text conditioning, the method uses score distillation sampling with a control factor from dense poses.

**Strengths:**

The paper tries to solve an important problem of current times. The strategy seems to be working and not counter intuitive. The results are also very encouraging.

**Weaknesses:**

The paper stitches together the existing methods and produces an intuitive pipeline used for other 3D asset creation from texts. Nerf with SMPL followed by Score distillation sampling seems to be very intuitive. Hence the novelty is a concern. The text prompts are also simple. The time taking to create the avatar is 2.5 hours which is too much of time. While the ablation shows favorable results but I am not sure why the coarse and fine stages are separately required?

**Questions:**

There are few important questions apart from the weaknesses which comes to my mind
1. The skinning is borrowed from SMPL which is coarse. The nearest vertices based weight can create a deformation problem.How that is not impacting?
2. What if we only deal with offset of SMPL instead of fine stage? Geometrically Fig 6 (a) is not showing significant advantage.
3. Its not clear why densepose is important than skeleton? Is it because skeleton is noisy? it is not clear from Fig 6(b) why the control net is behaving well with densepose. The argument is not strong. Cant there be other cases where densepose is failing?
4. We need more example and grounding for the need of dual space training.
5. How "p" is used in equation 4?
6.What is the comparison between CFG and CFG rescale trick?
7. Why Avatarverse has produced red belt instead of black belt in second row of Fig 3? has the comparison being done with proper negative prompt too?
8.It is not clear how the method is handing the janus problem.

---

> ### Author Response · Authors · 2023-11-22
> **Response to reviewer q3fv (1/4)**
>
> We thank the reviewer for the valuable feedback and agree that the results of our method are very encouraging. We respond to each of your comments one-by-one in what follows.
>
> > **Q1: The paper stitches together the existing methods and produces an intuitive pipeline used for other 3D asset creation from texts. Nerf with SMPL followed by Score distillation sampling seems to be very intuitive. Hence the novelty is a concern.**
>
> R1: Thank you for your feedback. We understand your perspective regarding the novelty of our work. While it's true that our method incorporates existing techniques, the process of combining them effectively to achieve high-quality results is a non-trivial task. We would like to highlight the following key points to clarify our contributions: 1) Previous methods, despite leveraging 3D representations and SDS for text-to-3D generation, are mainly focusing on static objects, which are not animatable. Differently, we propose a novel articulated DMTet representation for the more challenging 3D animatable human avatar creation. This novel representation, driven by SMPL-guided articulation, enables the creation of high-fidelity and animatable human avatars. 2) Optimizing the articulated 3D representation is not sufficient to achieve significant performance for animatable 3D avatar creation due to the lack of effective 2D pretrained guidance. To address this, we propose using a DensePose-conditional ControlNet to guide the human generation process, which provides a more accurate and dense description of a person's pose, thereby enhancing pose accuracy and alleviating the Janus problem. 3) To further improve the performance of our method, we incorporated a series of techniques into our framework, such as CFG rescale, part-aware super-resolution, coarse-to-fine optimization, and dual space training. 4) Our method achieves impressive stylized avatar creation results, as presented by **Reviewer B6UK**. Thus, we believe integrating all these techniques and demonstrating their effectiveness in animatable human avatar creation constitutes a valuable contribution, as also highlighted by **Reviewer L83u**.
>
> >**Q2: While the ablation shows favorable results but I am not sure why the coarse and fine stages are separately required?**
>
> R2: Thank you for your question regarding the necessity of our coarse-to-fine stages. The rationale behind this design choice is to address the limitations of each individual stage and to leverage their strengths in a complementary manner. The coarse NeRF stage struggles with rendering high-resolution images due to the significant computational cost of volume rendering, which limits its ability to learn detailed generation quality. On the other hand, the DMTet-only stage, while capable of achieving high-resolution rendering owing to its explicit mesh representation, is difficult to optimize from scratch and can result in geometry collapse (as shown in Fig. 6(a) of our ablation study). To address these issues, we levereage a coarse-to-fine pipeline. In the coarse stage, we use NeRF to train a coarse but reasonable result. This result is then used as an initialization to further optimize the articulated DMTet representation in the fine stage, achieving higher quality texture and geometry generation as well as avatar animation. We have provided more results from NeRF-only and DMTet-only methods on our [project page](https://avatarstudio3d.github.io/) (video grid with caption "Comparison between different 3D representations"). As can be seen, the NeRF-only generation results are relatively coarse and noisy geometry with floating artifacts can be observed, while the DMTet-only ones are not plausible. This further illustrates the necessity of our design choice.

---

> ### Author Response · Authors · 2023-11-22
> **Response to reviewer q3fv (2/4)**
>
> >**Q3: The skinning is borrowed from SMPL, which is coarse. The nearest vertices based weight can create a deformation problem. How is that not impacting?**
>
> R3: Thank you for your question regarding the impact of using SMPL for skinning and the potential deformation problems due to the nearest vertices based weight.
> 1. In this work, we use a relatively simple animation technique, i.e., SMPL-guided deformation. Despite its simplicity, it has proven effective for avatar animation, even for some avatars wearing loose clothing, as shown in the results ("A ballerina") on our [project page](https://avatarstudio3d.github.io/) (video grid with caption "More animation results"). We hypothesis that there are two main reasons why the potential deformation problems are not significantly impacting our results. First, we use a pre-trained 2D diffusion model to provide SDS guidance, which can correct areas where the SMPL deformation is inaccurate, leading to better animation results. Second, we use a strong DensePose conditioned guidance, which provides strong pose information, helping the model learn what the avatar should look like in different poses. These priors can help the model achieve plausible results on top of the coarse SMPL-guided animation.
> 2. Additionally, we have developed a non-rigid deformation module to learn residual deformation to compensate for areas where SMPL deformation is inaccurate. We found that adding such deformation makes the animation results more robust, especially for avatars with complex clothing, as shown on our [project page](https://avatarstudio3d.github.io/) (video grid with caption "More animation results-AvatarStudio with non-rigid deformation").
> Moving forward, we plan to explore more effective driving techniques and more robust driving signals to further improve the quality of our animations. We appreciate your insightful question and look forward to any further queries you may have.
>
> >**Q4: What if we only deal with offset of SMPL instead of fine stage? Geometrically Fig 6 (a) is not showing significant advantage.?**
>
> R4: Thank you for your question regarding the potential of using only the offset of SMPL without the fine stage.
> 1. Using NeRF-only representation without the fine stage presents two main issues. First, due to the memory consumption of volume rendering, NeRF-based representation cannot render high-resolution images, leading to coarse and low-resolution results with missing details. Second, the geometry results generated from the NeRF-based representation often contain noise and floating artifacts due to the lack of constraints, as shown on our [project page](https://avatarstudio3d.github.io/).
> 2. Conversely, in the fine stage, we use the coarse NeRF results as initialization to optimize our proposed articulated DMTet representation. This allows us to render higher resolution images for training, thereby achieving better texture and geometry generation results.
> We have added more video results of the comparison experiments on our [project page](https://avatarstudio3d.github.io/). As shown in these videos, adding our proposed articulated representation on top of the coarse nerf results, successfully generates textures and geometries with fine details, demonstrating its effectiveness.

---

> ### Author Response · Authors · 2023-11-22
> **Response to reviewer q3fv (3/4)**
>
> > **Q5: Its not clear why densepose is important than skeleton? Is it because skeleton is noisy? it is not clear from Fig 6(b) why the control net is behaving well with densepose. The argument is not strong. Cant there be other cases where densepose is failing?**
>
> R5: Thank you for your question regarding the importance of DensePose over the skeleton in our method.
> 1. Skeleton-based guidance is relatively sparse. the same skeleton-represented pose can potentially map to multiple real human poses, resulting in inaccuracies in the generated 3D avatars. The result in Fig 6 (b) demonstrates this. Moreover, this sparsity can also lead to ambiguity in distinguishing between frontal and back views, a phenomenon often referred to as the "Janus problem". On the other hand, DensePose provides a more detailed and accurate description of a person's pose. Its dense nature allows for a more precise mapping between the guidance and the actual human pose, thereby alleviating the aforementioned Janus problem and enhancing pose control accuracy. We have provided more results on our [project page](https://avatarstudio3d.github.io/) (video grid with caption "Comparison between different SDS guidances") to demonstrate this. Results based on skeleton guidance exhibit the Janus problem, while our proposed DensePose guidance method effectively mitigates this issue, yielding correct results.
>
>
> 2. However, we acknowledge that DensePose is not perfect. We have found that it is not precise enough for controlling avatar's hand creations, yielding artifacts. For example, in the supplementary materials for the "business woman" prompt, two wallets appear on the hands. A promising solution is to employ OpenPose hand model as guidance, which contains 21 finger joint keypoints, to model the fine-grained details of hands.
>
> >**Q6: It is not clear how the method is handing the janus problem.**
>
> R6: Thank you for your question regarding how our method handles the Janus problem. The Janus problem, a common issue in text-to-3D generative models, arises due to the lack of multi-view supervision. This problem often results in the model regenerating content described by the text prompt in different views, leading to multi-view inconsistency. To alleviate this issue, we render the SMPL mesh into DensePose conditions to guide the 3D human generation (see Fig. 6 (b)). This approach allows us to obtain precise view correspondence across views and between the 2D view and the 3D space. In contrast, skeleton conditions, which only have 18 keypoints (see Fig. 6 (b)), can only provide sparse correspondence between different views and from 2D view to 3D space. This sparsity often leads to difficulties in distinguishing between the front and back views of the generated human, resulting in problems such as multiple faces and unclear facial features in 3D avatar generation. We provide more comparison results on our [project page](https://avatarstudio3d.github.io/) (video grid with caption "Comparison between different SDS guidances"). For instance, in the case of the prompt "Albus Dumbledore", when we provide a back-view skeleton map projected from SMPL mesh as the conditional image and perform text-to-image generation, a frontal face appears in the generated image. This is because the keypoints reside in the SMPL mesh, and it is difficult to determine whether they are occluded, and thus guide the text-to-3D model to yield incorrect back-view image. In contrast, the DensePose control signals provide a more detailed and accurate description of a person's pose and view and thus guide the model to generate reasonable 3D avatar results, effectively mitigating the Janus problem. This has been demonstrated in the results on our project page. We hope this explanation addresses your question.
>
> > **Q7:  We need more example and grounding for the need of dual space training.**
>
> R7: Thank you for your question regarding the need for dual space training in our method. As suggested, we have added more video comparisons to illustrate the benefits of dual space training, which can be found on our [project page](https://avatarstudio3d.github.io/) (video grid with caption "Dual-space training"). We found that after using dual space training, the generated avatar shows a significant reduction in artifacts around the hip and shoulder areas. This demonstrates the effectiveness of adopting dual space training during the optimization process.

---

> ### Author Response · Authors · 2023-11-22
> **Response to reviewer q3fv (4/4)**
>
> >**Q8: The time taken to create the avatar is 2.5 hours, which is too much time.**
>
> R8: Thank you for your feedback regarding the optimization time of our method. We acknowledge that the time taken to create an avatar is a significant factor in the practicality of our approach. However, we would like to highlight the optimization time for our method is comparable to that of concurrent works in the field of text-to-3D generation. For example, methods such as Magic3D, AvatarVerse and ProlificDreamer require around 2-5 GPU hours with a single A100 GPU; while DreamAvatar and DreamWaltz require 2 hours and 3 hours on a single 2080Ti GPU and 3090 GPU, respectively. We have made efforts to reduce the optimization time of our method. By reducing the training steps (from 10,000 to 5,000 for the coarse stage, and from 3,000 to 2,000 for the fine stage), we have managed to decrease the optimization time from 2.5 hours to around 1 hour. The results, as shown on our [project page](https://avatarstudio3d.github.io/) (video grid with caption "Generations with fewer optimization steps"), are still comparable, demonstrating the effectiveness of our method and its potential to achieve more efficient results. In the future, we plan to explore other methods to further reduce optimization time, such as using more efficient Gaussian Splatting representation and stronger human prior and guidance.
>
> >**Q9: The text prompts are simple.**
>
> R9: Thank you for your feedback regarding the simplicity of our text prompts. We agree that testing our method with more complex prompts is essential for demonstrating its robustness and effectiveness. We have experimented with more complex prompts for human avatar creation, as shown on our [project page](https://avatarstudio3d.github.io/) (video grid with caption "Avatar creation with more complicated prompts"). Our method has shown promising results, effectively aligning the generated avatar with the detailed descriptions in the prompts. This demonstrates the robustness and effectiveness of our method even with complex prompts.
>
> > **Q10:  How p is used in equation 4?**
>
> R10: Thank you for your question regarding the use of "p" in Equation 4. In our method, "p" represents the parameters of the SMPL model. It is used to generate the DensePose control signal, denoted as $I_{cond}$. In Equation 4, $I_{cond}$ serves as the control signal provided to the DensePose ControlNet. This is used to calculate the SDS loss for gradient backpropagation and model parameters update. We acknowledge that this could have been clearer in our original manuscript and will clarify this point in our revision.
>
> > **Q11:  What is the comparison between CFG and CFG rescale trick?**
>
> R11: Thank you for your question regarding the comparison between the CFG and the CFG rescale trick. Current text-to-3D works use a large CFG scale when optimizing avatar representation with SDS to achieve reasonable results. However, a large CFG scale can produce severe color saturation, making the generated avatars look unreal. To alleviate the color saturation issue, we apply the CFG rescale trick from [1] to rescale x_t back to the original standard deviation w.r.t. a rescale strength $\phi$. Please refer to [1] for more details. This method effectively reduces color saturation and produces a more natural appearance in the generated avatar as shown in Fig.6(c). We will add a clearer introduction to the CFG rescale trick in revision.
>
> [1]  Shanchuan Lin, Bingchen Liu, Jiashi Li, and Xiao Yang. Common diffusion noise schedules and
> sample steps are flawed. arXiv, 2023b.
>
> > **Q12:  Why Avatarverse has produced red belt instead of black belt in second row of Fig 3? has the comparison being done with proper negative prompt too?**
>
> R12: Thanks for your comments. In the manuscript, we directly use the results from AvatarVerse paper for comparison. For a fair comparison, we also removed the negative prompt in our method. Despite removing the negative prompt, our results remained consistent, i.e., the color of the belt aligns with the text prompt. As for AvatarVerse's generation of a red belt given a black belt prompt, we hypothesize that this could be due to the training data bias in their ControlNet, leading to the generation of a red belt instead of a black one.

---

> > ### Comment · Reviewer_q3fv · 2023-11-22
> > **After Rebuttle**
> >
> > Many thanks to the author for the detailed and quality response. The coarse and fine stage is very clear now. But as authors also agrees, the method consists of many existing tools, primarily SMPL, DMTET along with standard SDS+Nerf combination. Therefore the pipeline becomes intuitive given the body of knowledge in the research community. This major concern still remains.

---

> ### Author Response · Authors · 2023-11-23
> **Further response to reviewer q3fv**
>
> We appreciate the continued feedback and the opportunity to further clarify the novelty of our contributions. While we leverage some existing techniques, our approach is far from being a simple combination of these methods. We introduced an articulated DMTet representation specifically designed for high-fidelity and animatable avatar creation, markedly distinct from previous text-to-3D and text-to-avatar works. Although the concept of an articulated DMTet might seem straightforward, effectively training this model to generate high-quality and animatable avatars is non-trivial. As demonstrated in Fig. 6 (a) of our paper and our [project page](https://avatarstudio3d.github.io/), an articulated DMTet model on its own often fails to generate avatars of sufficiently high quality.
>
> To tackle this challenge, we step-by-step analyzed and addressed the challenges in creating high-fidelity and animatable avatars. Specifically, we found that the model's initialization profoundly impacts its performance. Without proper initialization, the model struggles to yield promising results. To this end, we began by optimizing a coarse NeRF, and then used it to initialize our articulated DMTet model, which substantially alleviated learning difficulties and facilitated the generation process. Additionally, while we employed an SDS loss, our model diverges from prior text-to-3D work by integrating a dense human prior through a DensePose-conditional ControlNet. The DensePose-based ControlNet, as opposed to Stable Diffusion or Skeleton-based ControlNet, provides more detailed information about human pose and viewpoints, thereby enhancing the model's pose control capability and mitigating the Janus problem. The improvement can be observed on our [project page](https://avatarstudio3d.github.io/).
>
> Furthermore, to address issues with color saturation, we developed a CFG rescale method for text-to-avatar creation, which has not been explored in previous works. Finally, we integrated a series of techniques into our framework and conducted a comprehensive ablation study to evaluate their effectiveness.
>
> Through these improvements, our method achieved impressive avatar creation results, significantly outperforming all existing methods, as also mentioned by **Reviewer B6UK**. Therefore, we believe a coherent integration of all important modules and their impact, as demonstrated via a thorough ablation study, along with substantial performance improvements, are valuable contributions to the field, as also highlighted by **Reviewer L83u**. We hope our work could shed light on future research in this direction.

---

### Official Review · Reviewer_L83u · 2023-10-29

**Soundness:** 3 good
**Presentation:** 3 good
**Contribution:** 3 good
**Rating:** 8
**Confidence:** 4

**Summary:**

This paper studies the problem of text-guided 3D full-body human generation, and proposes several improvements over the previous SOTA method. Experiments demonstrate that the proposed method indeed improves the generation and animation quality, and all components are well-motivated in the ablation study.

**Strengths:**

- The paper is well-written and easy to follow
- The proposed method achieves SOTA results in text-to-3D avatar generation.
- The paper introduces several well-motivated techniques to improve the generation and animation quality, including using deep marching tetrahedra, densepose-guided ControlNet, part-based super-resolution, and SDS optimization in both canonical and deformed space. While some of these techniques have been used in other related tasks, demonstrating their effectiveness in this specific domain is a valuable contribution.

**Weaknesses:**

While one main focus of the paper is to improve animation, the animation still lacks realism. The animation is modeled via pure LBS with SMPL skinning weights and topology, thus cannot generate realistic non-linear cloth deformation, and cannot deal with loose clothing with other topologies such as skirts (skirts are split as shown in the animation results on the webpage).

**Questions:**

This paper proposes a set of simple yet effective techniques to improve text-to-avatar generation. The resulting method has demonstrated great quality improvement over previous SOTA. While the animation quality still needs further improvement, I believe this paper in its current state is already a valuable contribution to the field. I don't have any specific questions at this point.

---

> ### Author Response · Authors · 2023-11-22
> **Response to reviewer L83u**
>
> We thank the reviewer for the positive feedback and recognition that 1) our model achieves SOTA results in text-to-3D avatar generation;  2)  we introduce several well-motivated techniques to improve the generation and animation quality; 3) we demonstrate the effectiveness of each component in animatable avatar generation, which constitutes a valuable contribution; 4) the paper is well-written and easy to follow.
> We respond to each of your comments one-by-one in what follows.
>
> >**Q1: While one main focus of the paper is to improve animation, the animation still lacks realism.**
>
> Thank you for your insightful comments. We acknowledge the limitations you pointed out regarding the realism of our animation. The realism of our animation is influenced by the pre-trained models we use to generate or estimate the driving signals from text or video, which may inherently contain noise. This noise, when used to drive our avatars, can lead to a perceived lack of realism in the animation.
> While our current method struggles with certain types of clothing, such as skirts, we have found that it can handle loose clothing to a certain extent. As can be seen on our [project page](https://avatarstudio3d.github.io/) (video grid with caption "More animation results"), AvatarStudio can achieve plausible animation results on people wearing ballet costumes.
>
> Moreover, our current animation technique is based on a simple SMPL-guided articulation. We have build upon this by learning a stronger non-rigid deformation network to predict residual deformation to handle loose cloth modeling and alleviate inaccurate SMPL deformation. We have achieved promising results with more challenging clothes, e.g., skirts and dresses, by incorporating this approach, as shown on our project page (video grid with caption "More animation results-AvatarStudio with non-rigid deformation"). Moving forward, we plan to explore more effective driving techniques and more robust driving signals to further improve the quality of our animations.

---

### Official Review · Reviewer_B6UK · 2023-10-30

**Soundness:** 3 good
**Presentation:** 3 good
**Contribution:** 2 fair
**Rating:** 5
**Confidence:** 4

**Summary:**

This paper presents a new method for generating 3D human avatars from text prompts. To this end, the authors propose to generate 3d avatars in a coarse-to-fine manner. The coarse stage is based on NeRF, while the fine stage takes the coarse result as initialization and refines it with explicit mesh-based representation. In addition, the authors propose to perform SDS sampling with a diffusion model conditioned on DensePose, which allows for better view consitency. Experiments show that the proposed method is able to generate animatable human avatars from only text input.

**Strengths:**

* The proposed method is able to generate high-quality human avatars from only text input, and the generated avatars have clear appearance details. Experiments show that the proposed method outperforms existing pipelines. Moreover, the authors also demonstrate stylized avatar creation given a style image as an additional condition, which is very impressive.

* The authors propose to using DensePose-conditioned ControlNet for SDS supervision. Experiments show that it can achieves precise and stable pose control, which may inspire future work on avatar generation or other catogery-specific 3D generation tasks.

* The paper is well-writen and easy to follow.

**Weaknesses:**

* In Abstract and Introduction, the authors claim that using ControlNet conditioned on DensePose offers a benefit on view consitency, but I cannot find any experiments to support this claim. In Figure 6(b), the authors conduct an ablation study to evaluate the effects of different SDS supervision, but the results only show that leveraging skeleton-based ControlNet may suffer from leg pose error. Existing methods like DreamHuman and DreamAvatar are typically based on original Stable Diffusion or skeleton-based ControlNet, and I didn't notice any view inconsistency issues.

* Generating 3D models in a coarse-to-fine manner with two representations is not a new idea. In fact, it has already been proposed in ProlificDreamer [Wang et al, 2023], which also leverage NeRF for initialization and uses DMTet for further refinement. Therefore, I don't think it can be regarded as a technical contribution.

* Jointly optimizing the textured avatar mesh in both deformed and canonical spaces is also not new and has been proposed in DreamAvatar, which also proposes to perform SDS supervision in both canonical space and posed space.

* Overall, although this paper demonstrates good results, its technical novelty is not strong enough to me. I feel that the proposed method is more like a combination of existing techniques and tricks.

**Questions:**

See [Weaknesses]

---

> ### Author Response · Authors · 2023-11-22
> **Response to reviewer B6UK (1/2)**
>
> We thank the reviewer for the positive feedback and recognition that 1) the proposed method outperforms existing pipeline and the stylized avatar creation results are very impressive; 2) the DensePose-conditioned ControlNet for SDS supervision can achieves precise and stable pose control, which may inspire future work on avatar generation or other catogery-specific 3D generation tasks; 3) the paper is well-written and easy to follow.
> We respond to each of your comments one-by-one in what follows.
>
> >**Q1: Lack of experiments to support this claim that using ControlNet conditioned on DensePose offers a benefit on view consistency.**
>
> R1:  Thanks for your comments. In our paper, we posit that leveraging DensePose as a form of SDS guidance for 3D generation offers a significant advantage over keypoint or skeleton-based guidance. The reason is as follows: 1) Skeleton-based guidance, while effective in many scenarios, can be relatively sparse. This sparsity can lead to ambiguity in distinguishing between frontal and back views, a phenomenon often referred to as the "Janus problem". Besides, due to its sparsity, the same skeleton-represented pose can potentially map to multiple real human poses, leading to inaccuracies in the generated 3D avatars. 2) On the other hand, DensePose provides a more detailed and accurate description of a person's pose. Its dense nature allows for a more precise mapping between the guidance and the actual human pose, thereby alleviating the aforementioned Janus problem and enhancing view consistency.
>  To support our claim, we visualize more qualitative results of 3D avatars generated using Stable Diffusion, skeleton-conditional and DensePose-conditional guidances on our [project page](https://avatarstudio3d.github.io/) (video grid with caption "Comparison between different SDS guidances"). As shown in the videos, the models based on the Stable Diffusion and the skeleton-conditional ControlNet generate a frontal face in the back view. In contrast, using DensePose-conditional ControlNet as guidance, the model can generate plausible 3D avatar results both in the front and the back views.
>
> >**Q2:  Generating 3D models in a coarse-to-fine manner with two representations is not a new idea.**
>
> R2: Thank you for your valuable feedback. We acknowledge that the concept of generating 3D models in a coarse-to-fine manner is not new, and we did not intend to claim this as our novelty. In the revision, we will tune down our claim to be more in line with your comments. However, we believe that our method introduces significant advancements and unique contributions in the context of animatable human avatar generation, which we would like to elaborate on:
> 1. While previous methods have explored the coarse-to-fine pipeline for 3D generation, they mainly focus on static objects, which are not animatable. In contrast, our method extends this coarse-to-fine paradigm to animatable human avatar generation, which is a more challenging task. This is a significant step forward as it allows for more dynamic 3D modeling. In our refinement stage, unlike ProlificDreamer that uses DMTet for image quality enhancement on static objects, we propose an articulated DMTet representation. This novel representation, driven by SMPL-guided articulation, enables the creation of animatable human avatars, a feature not present in previous text-to-3D works.
> 2. Having established this novel representation, we found that it alone was not sufficient to achieve optimal performance for animatable 3D avatar generation. Therefore, we introduced several improvements to enhance the efficacy of our approach. Specifically, we propose the use of DensePose-conditioned ControlNet to guide the human generation process. This leverages the DensePose human prior as multi-view supervision, providing a more accurate and dense description of a person's pose, thereby enhancing pose accuracy and alleviating the Janus problem. To further improve the performance of our method, we incorporated a series of techniques into our framework, such as CFG rescale, part-aware super-resolution, coarse-to-fine optimization, and dual space training. These techniques, when combined, effectively enhance our method and enable us to achieve plausible results.

---

> ### Author Response · Authors · 2023-11-22
> **Response to reviewer B6UK (2/2)**
>
> >**Q3:  Jointly optimizing the textured avatar mesh in both deformed and canonical spaces is also not new and has been proposed in DreamAvatar.**
>
> R3: Thanks for pointing out this issue. We acknowledge that the concept of performing SDS supervision in both spaces has been previously proposed in works such as DreamAvatar. We will add references in the revision. Also, dual-space training is not intended to be claimed as a novelty of this work. Rather, we adopted it as a technique to improve the accuracy of animation. Our main contributions lie elsewhere, specifically in the development of an articulated DMTet representation for animatable human avatar generation, the introduction of DensePose-conditional ControlNet guidance, and the incorporation of techniques such as CFG rescale, part-aware SR, dual space training, and coarse-to-fine optimization. These methods, when combined, have allowed us to significantly advance the state of the art in animatable 3D avatar generation. We hope this response addresses your concerns and we appreciate your feedback as it helps us improve the clarity of our work.
>
> > **Q4: Overall, although this paper demonstrates good results, its technical novelty is not strong enough to me.**
>
> R4: Thank you for your feedback. We understand your perspective regarding the novelty of our work. While it's true that our method incorporates existing techniques, the process of combining them effectively to achieve high-quality results is a non-trivial task. We would like to highlight the following key points to clarify our contributions: 1) we propose a novel articulated DMTet representation for the challenging 3D animatable human avatar creation. This novel representation, driven by SMPL-guided articulation, enables the creation of high-fidelity and animatable human avatars, a feature not present in previous text-to-3D works. 2) Optimizing the articulated 3D representation is not sufficient to achieve significant performance for animatable 3D avatar creation due to the lack of effective 2D pretrained guidance. To address this, we propose using a DensePose-conditional ControlNet to guide the human generation process, which provides a more accurate and dense description of a person's pose, thereby enhancing pose accuracy and alleviating the Janus problem. 3) To further improve the performance of our method, we incorporated a series of techniques into our framework, such as CFG rescale, part-aware super-resolution, coarse-to-fine optimization, and dual space training. 4) Our method achieves promising avatar animation and stylized avatar creation. Thus, we believe integrating all these techniques and demonstrating their effectiveness in animatable human avatar creation constitutes a valuable contribution, as also highlighted by **Reviewer L83u**.

---

### Author Response · Authors · 2023-11-23

Dear reviewers,

Thank you for your effort in reviewing our paper and offering valuable comments. We have provided corresponding responses, which we believe have covered your concerns. As the discussion period is close to its end, should further queries arise, we are fully prepared to answer. Your insights are invaluable to us. Thank you sincerely for your time and support!

---

### Meta-Review · Area_Chair_x7sB · 2023-12-10

**Metareview:**

The paper introduces a novel method for generating 3D human avatars from text prompts. This paper has been rigorously reviewed by three experts in the field, leading to a set of mixed reviews. The reviewers have positively noted the ability of the proposed method to generate high-quality human avatars based solely on text input. They acknowledge the clarity of appearance details in the generated avatars.

However, despite these strengths, there remain pivotal concerns that preclude a recommendation for acceptance at this stage. The reviewers point out that the framework mainly leverages existing and well-established techniques, such as DMTET and the standard SDS+Nerf combination. This reliance on well-known practices raises questions about the novelty and technical contributions over the current state of the arts.

While the paper is undoubtedly meritorious in its ability to produce detailed avatars, the decision, considering the reviewers' feedback, leans towards not recommending acceptance in its present form. We encourage the authors to take the reviewers' feedback into consideration. Particularly, there is a need to highlight and enhance the novel aspects of the proposed method or to combine the existing techniques in a more innovative manner that clearly demonstrates a significant advancement in the field.

**Justification For Why Not Higher Score:**

The proposed framework mainly consists of existing and well-known techniques, e.g., DMTET along with standard SDS+Nerf combination.

**Justification For Why Not Lower Score:**

N/A

---

### Decision · Program_Chairs · 2024-01-16

Reject